# Placental histopathology in preterm birth with confirmed maternal infection: A systematic literature review

Aung Myat Min[1]*, Makoto Saito[2], Julie A. Simpson[3], Stephen H. Kennedy[4,5], François H. Nosten[1,6], Rose McGready[1,6]

1 Shoklo Malaria Research Unit, Mahidol-Oxford Tropical Medicine Research Unit, Faculty of Tropical Medicine, Mahidol University, Mae Sot, Tak, Thailand, 2 Division of Infectious Diseases, Advanced Clinical Research Center, Institute of Medical Science, University of Tokyo, Tokyo, Japan, 3 Centre for Epidemiology and Biostatistics, Melbourne School of Population and Global Health, University of Melbourne, Melbourne, Australia, 4 Nuffield Department of Women's & Reproductive Health, University of Oxford, Oxford, United Kingdom, 5 Oxford Maternal & Perinatal Health Institute, Green Templeton College, University of Oxford, Oxford, United Kingdom, 6 Centre for Tropical Medicine and Global Health, Nuffield Department of Medicine, University of Oxford, Oxford, United Kingdom

* ayemin@shoklo-unit.com

**Data Availability Statement:** All relevant data are within the manuscript and its Supporting information files.

## Abstract

Four in five neonatal deaths of preterm births occur in low and middle income countries and placental histopathology examination can help clarify the pathogenesis. Infection is known to play a significant role in preterm birth. The aim of this systematic review is to explore the association between placental histopathological abnormality and preterm birth in the presence of confirmed infection. PubMed/Medline, Scopus, Web of Science and Embase were searched using the keywords related to preterm birth, placental histopathology and infection. Titles and abstracts were screened and the full texts of eligible articles were reviewed to extract and summarise data. Of 1529 articles, only 23 studies (13 bacterial, 6 viral and 4 parasitic) were included, and they used 7 different gestational age windows, and 20 different histopathological classification systems, precluding data pooling. Despite this, histopathological chorioamnionitis, and funisitis (when examined) were commonly observed in preterm birth complicated by confirmed bacterial or viral, but not parasitic, infection. The presence of malaria parasites but not pigment in placenta was reported to increase the risk of PTB, but this finding was inconclusive. One in three studies were conducted in low and middle income countries. An array of: definitions of preterm birth subgroups, histological classification systems, histopathologic abnormalities and diagnostic methods to identify infections were reported in this systematic review. Commitment to using standardised terminology and classification of histopathological abnormalities associated with infections is needed to identify causality and potential treatment of preterm birth. Studies on preterm birth needs to occur in high burden countries and control for clinical characteristics (maternal, fetal, labor, and placental) that may have an impact on placental histopathological abnormalities.

**Funding:** This research was funded in whole, or in part, by the Wellcome Trust [220211]. For the purpose of Open Access, the author has applied a CC BY public copyright license to any Author Accepted Manuscript version arising from this submission.

**Competing interests:** The authors have declared that no competing interests exist.

## Introduction

Preterm birth (PTB), occurring before 37 completed weeks' gestation, is recognised globally as the main cause of neonatal mortality and morbidity [1, 2]. In 2014, an estimated 14.8 million babies (1 in 10), were born preterm; of these, 80% were born in Asia and sub-Saharan Africa [3]. However, accurate estimation of PTB rates, especially in low- and middle-income countries (LMICs) is often difficult because of the unavailability of ultrasonography for early and precise dating of the pregnancy [4].

PTB can be classified as: 1) spontaneous PTB with intact membranes, 2) preterm premature/pre-labour rupture of membranes (PPROM) or 3) indicated or induced due to maternal and/or fetal complications [5]. PTB is further categorised in the following subgroups: late preterm ($\geq$32 to <37 weeks' gestation), very preterm (28 to <32 weeks' gestation) and extremely preterm (<28 weeks' gestation), with the last group having the highest mortality and morbidity [6]. The associated morbidity includes a wide range of conditions with long-term impairment of motor- and neurocognitive development. Preterm neonates in high-income countries (HICs) have a much better chance of survival than those in LMICs [4, 7]. In addition, the support services to care for related sequelae such as visual, hearing or learning difficulties, frequently do not exist in LMIC settings [8].

The placenta plays a key role in creating the optimal environment required to sustain a pregnancy until term, which is crucial for optimal neonatal outcomes. Changes in placental microanatomy can lead to severe consequences (e.g., PTB); hence, historically, histopathological examination has been an important method to understand the possible cause of PTB [9]. The placental inflammatory changes detected by histopathological examination are usually classified into acute or chronic inflammatory responses of the placenta, which can be further divided into maternal inflammatory response (chorioamnionitis) and fetal inflammatory response (umbilical vasculitis) [10]. Chorioamnionitis can be graded histologically by severity, chronicity and by fetal inflammatory responses that are observed in the chorionic plate and umbilical cord. Unfortunately, however, there is no consensus on placental sampling, tissue processing methods, or the recording and reporting of histopathological results [11]. The need for standardisation arises because common placental histopathological lesions are seen in PTB cases with and without infection. These include: villitis of unknown origin (destructive villous inflammatory lesion), acute or chronic chorioamnionitis (inflammation of chorioamnionic membrane), and chronic deciduitis (inflammation of decidua basalis) [12–15].

Placental histopathological abnormalities from intra-amniotic infection can be caused by "1) ascending bacterial infection from lower female reproductive tract (major pathway), 2) haematogenous spread of a bacterial, viral or parasitic infection through the placenta, 3) invasive procedures like amniocentesis, and 4) retrograde spread via the fallopian tubes, which is less likely to occur" [5]. The risk of spontaneous PTB also depends on the local vaginal microbial flora: microbiome studies have shown that elevated *Ureaplasma or Gardnerella* levels increase the risk of spontaneous PTB [5, 16, 17], while a vaginal flora rich in *Lactobacilli* is associated with a reduced spontaneous PTB risk [18]. Systemic infection in pregnancy, in particular infection of the urogenital tract and severe viral and malarial infections, are causally associated with PTB. Most notably, microbial genital tract infections are responsible for up to 25–40% of PTBs [19]. While the acute inflammatory response mostly occurs with ascending infection due to bacterial and fungal infections, haematological spread of viral and parasitic infections are more commonly a cause of chronic inflammatory response of the placenta resulting in chronic chorioamnionitis and chronic villitis [20, 21]. While intervillous inflammatory infiltrates by mononuclear cells are associated with acute and chronic malaria infection [22], histopathological findings of malaria infection are more frequently characterised by the

presence of malaria parasites and/or pigment in the intervillous space of the placenta but not histologic chorioamnionitis or funisitis [22, 23].

Placental histopathological abnormalities can also arise due to PPROM and the length of time elapsed between membrane rupture and birth, which can make it difficult to differentiate PTB phenotypes based on the presence or absence of intrauterine acute inflammation/infection, immunologically mediated processes and uterine ischaemia [24]. This systematic literature review explores: (1) the association between placental histopathological abnormalities and PTB in the presence of confirmed bacterial, viral or parasitic infection, and (2) the possible association between the identified microorganism and specific placental histopathological abnormalities.

## Materials and methods

The rationale, objective and search strategy of this systematic review were registered in the International Prospective Registers of Systematic Reviews (PROSPERO) under the registration number CRD42019137099 [25].

### Search strategy

A systematic literature review following PRISMA guidelines [26] was conducted to identify studies reporting placental histopathology in PTB with a confirmed infection. Four databases: (PubMed/Medline, Scopus, Web of Science and Embase) were searched. The original search was conducted in March 2019 and continually updated until February 2021 focusing on three components: placental histopathology, PTB and infection. The search terms (S1 Table) included: "pregnan* AND (preterm* OR premature*) AND placenta*AND (histopatholog* OR histology* patholog*) AND (Infect* OR microorganism OR bacteria* OR virus* OR parasite*)". No temporal, geographical or language restrictions were applied to the search.

Interventional and observational studies were included if the histopathological findings were related to PTB and any type of infection (bacterial, viral or parasitic). The inclusion criteria used for full text screening were English language, full text availability, original research describing PTB/PPROM, placental histopathology assessed, micro-organisms reported, and most importantly that the study described the association between placental histopathology and infection in PTB/PPROM. In the event of discrepancies during the selection process, a second assessment was conducted to resolve the inconsistency. The bibliographic management software, EndNote (version X9.3.3), was used to organise the articles.

Studies that did not unambiguously include the three components in triangulation, i.e. histopathological examination, infection and PTB, were excluded. Histopathological placental findings reported in isolation or as part of a series of investigations were included if the other conditions were satisfied. Systematic reviews on histopathology and PTB were checked for any possible articles that may have been missed by the primary search.

After inclusion, the following data were extracted: year, country, economic country profile (HIC or LMIC), study design, timing of inclusion of participants (prospective at time of infection or at birth), estimated gestational age (EGA) of included PTBs, method of gestational age assessment, presence of rupture of membranes (ROM), sample size, histopathological classification and reporting system, confirmation of infection, and methodology and blinding of pathologist. Reference to the histopathological abnormality classification system was extracted when provided. High numbers of papers were excluded because they compared only two of three inclusion criteria in their analysis: placental histopathology and PTB, or microbial infection and histopathology, or microbial infection and PTB.

The methodological quality of the studies was assessed by evaluating study design, inclusion criteria and the blinding of investigators to pregnancy outcomes or causation. To address the risk of bias, the Newcastle-Ottawa quality assessment scale for case-control and cohort studies was used [27]. Two independent review authors (AMM and RM) appraised the selected studies; in the event of a discrepancy, a third reviewer (FHN) was consulted. Any encountered quality issues or concerns were addressed by referring to the particular section and grading the severity of the bias (moderate, serious or critical). Twenty selected clinical characteristics (maternal, fetal, delivery and infection) associated with PTB were extracted and a binary score was applied for presence or absence.

### Subgroup allocation

As "infection" encompasses a broad range of causative pathogens and studies typically had a narrow range of interest, the term "infection" was divided into three groups according to the following infectious agents: bacteria, viruses or parasites.

## Results

### Study characteristics

A total of 3277 records were identified through database searching. After removing duplicates, 1529 articles were screened against the defined eligibility criteria. Of these, 1407 were excluded after screening the study title and abstract: animal studies = 135; case reports, case series or review articles = 495; articles that did not report confirmed infection = 164; articles focused on microbiome, proteomic or serum markers = 366; multiple pregnancies = 13; not preterm = 91; placental histopathology not reported = 143. This left only 23 studies for inclusion (bacteria, viral and parasitic) (Fig 1) [28–50].

Of included research articles that were published between 1988 and 2020, 15/23 studies (65.2%) were reported in the last 10 years. Only 8 (34.8%) were conducted in LMIC settings (Fig 2).

The study characteristics, PTB definition, placental histopathology classification system, range of infective organisms, and sample size are summarized in Table 1. The majority (56.5%, 13/23) of studies examined the association between bacterial infection and placental histopathology in PTB [30, 31, 35–39, 41, 44, 46–49], six studies (26%) examined viral infection [29, 32, 33, 42, 45, 50], and four studies (17%) examined studied parasites [28, 34, 40, 43].

Most studies mentioned or clarified only a few of the clinical characteristics (maternal, fetal, delivery and infection) associated with PTB: median score of 6 [range 1–12] out of a possible total of 20 (S2 Table).

### Quality assessment

Of the included studies, 13 were cohort studies, seven were case-control studies and three were sub-analysis of randomised controlled trials. Most included studies (65%, 15/23) were conducted prospectively and 14 mentioned the blinding of the pathologist(s) to clinical data [29–31, 34–37, 39, 40, 43–45, 47, 50] (Table 1). The gestational age assessment by ultrasound was clearly specified in 12 articles [28, 29, 31, 36, 37, 39, 40, 42, 43, 45, 47, 50]. The risk of bias was low using the Newcastle-Ottawa quality assessment scale: median 6 (maximum score 7) for cohort studies and median 7 (maximum 8) for case-control studies (S3 Table).

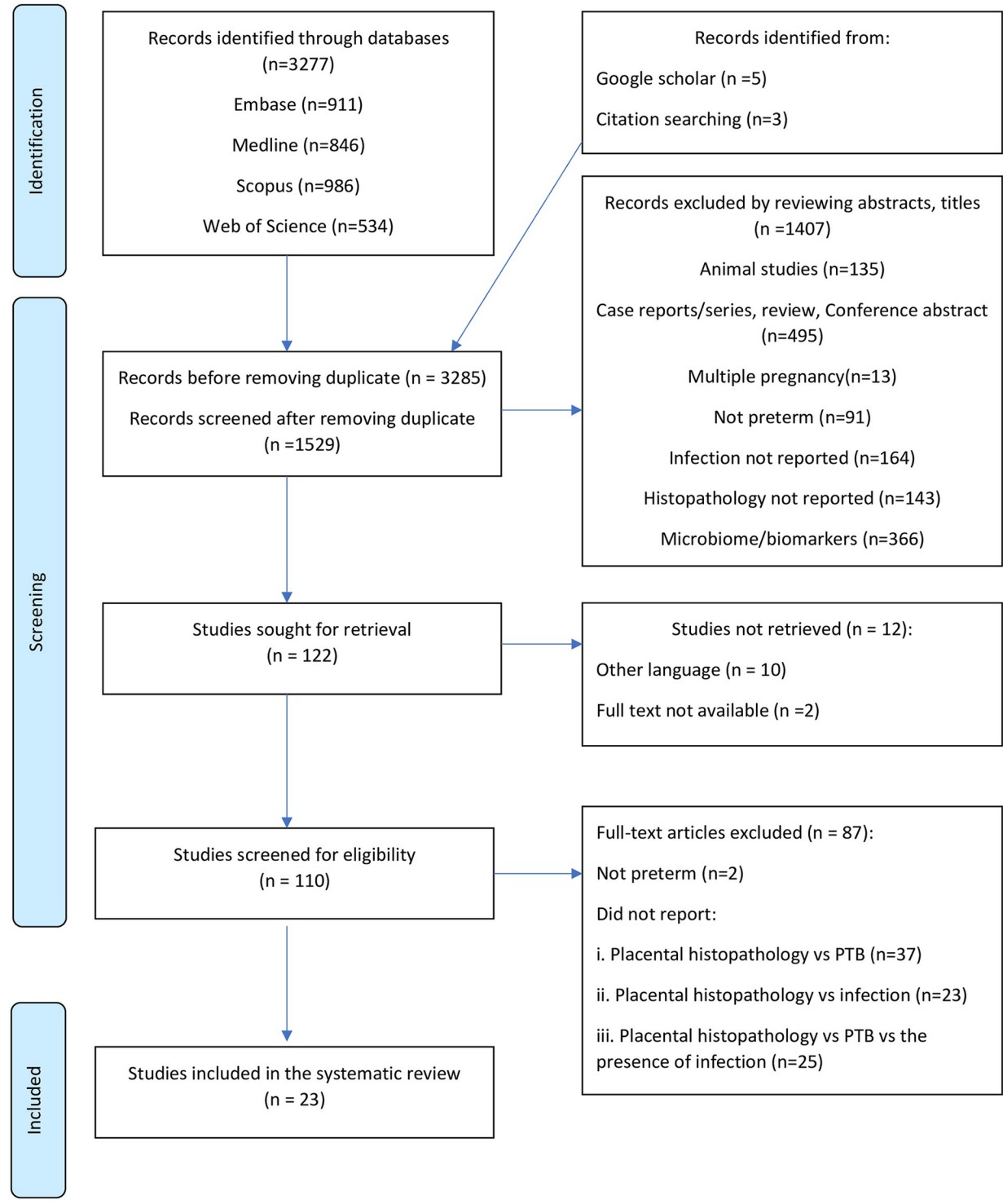

**Fig 1. Prisma flow chart.**

## Variations of study participants

While an estimated gestational age of less than 37 weeks was the most common PTB definition, various lower cut-offs were used: seven different gestational age windows were described

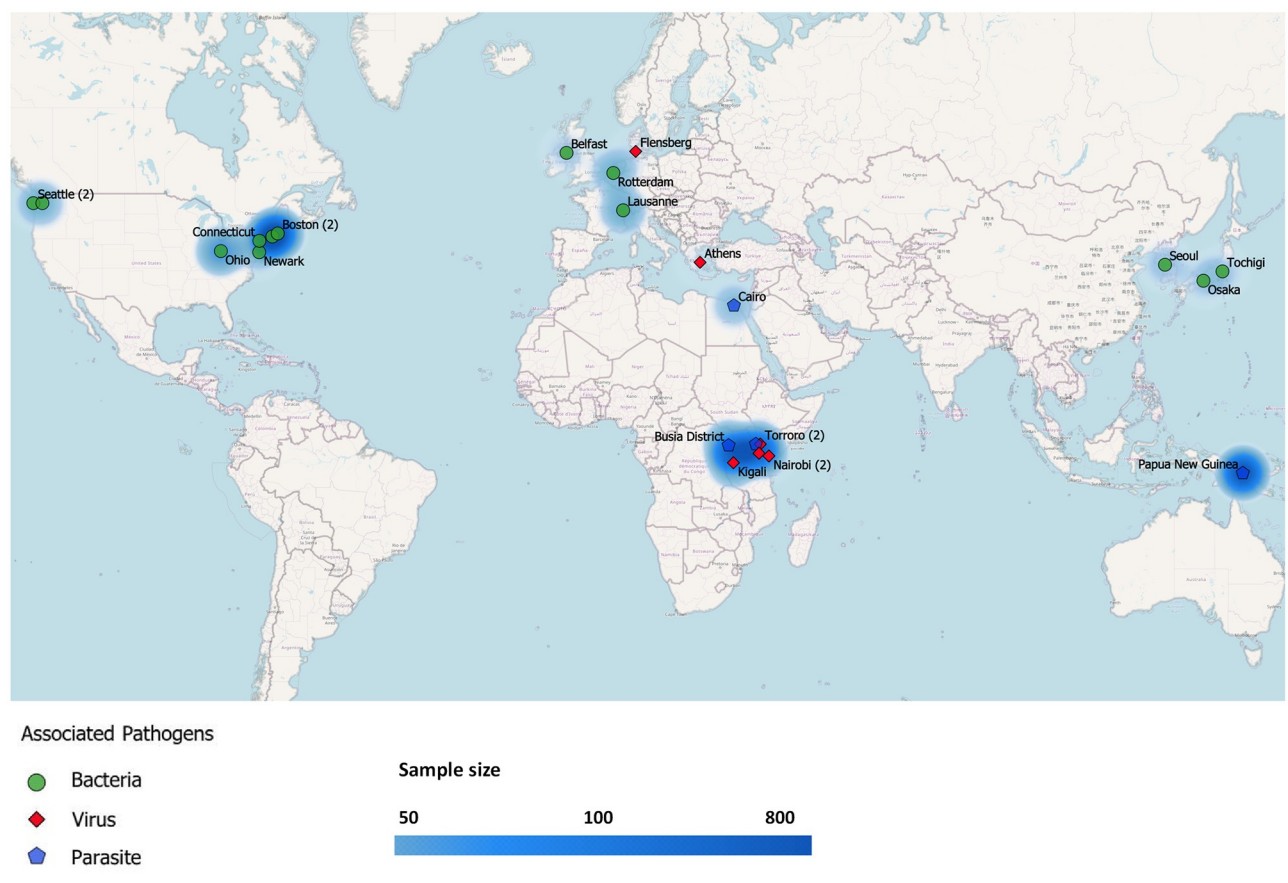

**Fig 2. Map of countries where studies were conducted.**

in 23 studies. Moreover, eight studies recruited early and extremely preterm newborns as the study participants and the classification of PTB was not uniform [30, 31, 35, 37–39, 44, 46]. PTB subcategories (spontaneous PTB with intact membranes, PPROM and indicated or induced due to maternal or fetal complications) were frequently not mentioned (S2 Table). Only 11 studies were designed to investigate the associations of placental histological abnormalities and infections in PTB [30, 31, 33, 35–37, 39, 41, 44, 46, 47].

## Histopathological methods

Placental tissue processing techniques varied and 21 different publications [10, 22, 23, 51–68] were used to support histopathological classification systems, which were reported in the included studies (S4 Table). Six studies used standardised histopathological definitions: four [29, 35, 39, 49] using Redline *et al.* 2003 [10] and two using Khong *et al.* [59]. Seven studies used their own definitions of histological chorioamnionitis [36–38, 41, 42, 47, 48, 50] and three malaria studies [28, 40, 43] used four classifications [61]. Evidence of reporting of the severity of placental histopathological abnormality was limited to eight (34.8%) of studies [28, 32, 35, 39, 44, 47–49].

## Pathogen detection methods

Culture technique was used in 11 bacterial studies [31, 35–38, 41, 44, 46–49] and molecular methods were performed in three studies to identify the infective bacterial organisms [30, 39,

**Table 1. Study characteristics.**

| Author name, Year[*] | Type of infection | PTB definitionused (weeks) | Type of study | Retrospective/ Prospective | No. PTB (%) | Country | GNI[**] per capita | Blinding of pathologist |
|---|---|---|---|---|---|---|---|---|
| **Cox, 2016** [30] | Bacteria | 13–37 | Case Control | Retrospective | 57/57 (100%) | Ireland | HIC | Yes |
| **Dammann, 2003** [31] | Bacteria | Not clearly defined | Case Control | Retrospective | 464/464 (100%) | USA | HIC | Yes |
| **Hecht, 2008** [35] | Bacteria | <28 | Cohort | Retrospective | 835/835 (100%) | USA | HIC | Yes |
| **Hillier, 1988** [37] | Bacteria | <37 | Case control | Retrospective | 38/74 (51%) | USA | HIC | Yes |
| **Hillier, 1991** [36] | Bacteria | <34 | Case control | Retrospective | 112/268 (42%) | USA | HIC | Yes |
| **Honma, 2007** [38] | Bacteria | <32 | Cohort | Retrospective | 105/105 (100%) | Japan | HIC | Not mentioned |
| **Ingrid, 2011** [39] | Bacteria | <32 | Cohort | Prospective | 304/304 (100%) | Netherland | HIC | Yes |
| **Kwak, 2014** [41] | Bacteria | <37 | Cohort | Prospective | 179/179 (100%) | Korea | HIC | Not mentioned |
| **Namba, 2010** [44] | Bacteria | <32 | Cohort | Prospective | 151/151 (100%) | Japan | HIC | Yes |
| **Patel, 2018** [46] | Bacteria | 23–33+6 | Cohort | Retrospective | 181/181 (100%) | USA | HIC | Not mentioned |
| **Pettker, 2007** [47] | Bacteria | <37 | Case Control | Prospective | 183/183 (100%) | USA | HIC | Yes |
| **Queiros da Mota, 2013** [48] | Bacteria | <37 | Cohort | Prospective | 202/376 (53.7%) | Switzerland | HIC | Not mentioned |
| **Sweeney 2016** [49] | Bacteria | <37 | Cohort | Prospective | 535/535 (100%) | USA | HIC | Not mentioned |
| **Ategeka, 2019** [29] | Virus | <37 | Sub-analysis of RCT | Prospective | 18/191 (9.4%) | Torroro | LMIC | Yes |
| **Feist, 2020** [32] | Virus | <37 | Cohort | Retrospective | 14/50 (28%) | Germany | HIC | Not mentioned |
| **Gichangi, 1993** [33] | Virus | <37 | Case control | Prospective | 117/467 (31.4%) | Kenya | LMIC | Not mentioned |
| **Ladner, 1998** [42] | Virus | <37 | Cohort | Prospective | 39/275 (14%) | Rwanda | LMIC | Not mentioned |
| **Ombimbo, 2019** [45] | Virus | <37 | Cohort | Prospective | 81/101 (80.2%) | Kenya | LMIC | Yes |
| **Tsekoura, 2010** [50] | Virus | <37 | Case Control | Prospective | 37/58 (63.8%) | Greece | HIC | Yes |
| **Ategeka, 2020** [28] | Parasite | <37 | Sub-analysis of RCT | Prospective | 37/637 (5.8%) | Uganda | LMIC | Not mentioned |
| **Kapisi, 2017** [40] | Parasite | <37 | Sub-analysis of RCT | Prospective | 26/282 (9.2%) | Uganda | LMIC | Yes |
| **Lufele, 2017** [43] | Parasite | <37 | Cohort | Prospective | 81/962 (8.4%) | Papua New Guinea | LMIC | Yes |
| **Saad, 2017** [34] | Parasite | <37 | Cohort | Prospective | 37/240 (15.4%) | Egypt | LMIC | Yes |

Abbreviations: GNI, gross national income; HIC, high-income country; HIV, Human immunodeficiency virus; LMIC low- to middle-income country; RCT, randomised controlled trial

[*]year of publication.

[**]gross national income based on World Bank data. https://data.worldbank.org/indicator/NY.GNP.PCAP.CD?locations=TH.

49]. The majority of studies used a single sampling method (placental tissue or placental swab or vaginal swab) while three studies used two or more sampling techniques [37, 46, 47]. For studies targeting human immunodeficiency virus (HIV) infection, two studies used an enzyme linked immunosorbent assay (ELISA) as a diagnostic test [33, 42], while another two studies did not mention the diagnostic methods [29, 45]. In another two studies that targeted viral infection, polymerase chain reaction (PCR) was performed to detect adenovirus, human papilloma virus (HPV) and enterovirus [32, 50]. Two of three malaria studies confirmed infection by peripheral blood films, placental blood microscopy, and Loop-mediated isothermal application (LAMP) [28, 40] but the histopathological detection of malaria parasites and pigment in placenta was used in all three malaria studies [28, 40, 43].

## Placental histopathologic findings in studies of preterm birth with bacterial infection

There were 13 studies that reported placental histopathology and PTB with bacterial infection. A wide range of bacterial species (n = 54) were reported with *Ureaplasma* species (7/13), Group B *streptococcus* (5/13) and *Mycoplasma hominis* (6/13) being the most common. Four articles specifically investigated only *Ureaplasma* and *Mycoplasma* species [31, 38, 41, 44]. Another two studies examined the association between selected microorganisms (*Chlamydia trachomatis* and group B *Streptococcus* (*GBS*)) and PTB [39, 46]. Histological chorioamnionitis was the most common placental histopathological abnormality in PTB with bacterial infection [29–31, 33, 35–39, 41, 42, 46–49]. Four studies reported both chorioamnionitis and funisitis [31, 33, 35, 47].

Ingrid *et al*. used PCR to identify *C. trachomatis* and studied its association with placental histopathology in PTB. They reported that a chlamydia infection was two times as likely to be associated with placental histological inflammation in preterm cases compared to absence of chlamydia infection [39]. The rates of histological chorioamnionitis between *GBS* positive and negative groups of PPROM were not significantly different, as described by Patel *et al*. [46]. Pettket *et al*. reported that in preterm cases with a positive amniotic fluid culture, acute histological chorioamnionitis and funisitis were more pronounced when compared to preterm amniotic fluid culture negative cases [47]. Querios da Mota *et al*. showed that the proportion of histological chorioamnionitis with positive bacteriological culture in preterm placentas was 25% and more than 50% of microbial culture results came from preterm placentas [48]. Sweeney *et al*. also concluded that histological chorioamnionitis was more likely to be associated with moderate/late PTB, complicated by an infection with *Ureaplasma* species, the most common organisms identified in placental culture [49]. More detailed information on these studies is provided in Table 2.

## Placental histopathology in studies of preterm birth with viral infections

The main placental abnormality in four of six viral infection studies was histologically confirmed chorioamnionitis [29, 33, 42, 50]. Ategeka *et al*. reported that the risk of PTB was significantly increased with histological evidence of severe "maternal acute chorioamnionitis" in HIV infected pregnancies but not associated with "fetal acute chorioamnionitis" (maternal and fetal acute chorioamnionitis refers to maternal & fetal acute inflammatory response to infection in this article) [29]. Obimbo *et al*. reported that fibrinoid deposition with villous degeneration, syncytiotrophoblast delamination, increased red blood cell adhesion to terminal villi and increased number of capillaries were significantly associated with preterm placentas in HIV infection and/or antiretroviral therapy [45]. Gichangi *et al*. reported that the likelihood of having moderate to severe histological chorioamnionitis and moderate to severe funisitis

**Table 2. Identified bacterial infection and histopathological changes in preterm birth.**

| Author, Year | Reported pathogenic microorganism | Diagnostic method | No. PTB (%) | Infection | Histopathological abnormalities | Triad of histopathology, infection and PTB |
|---|---|---|---|---|---|---|
| Cox, 2016 [30] | *Mycoplasma, Ureaplasma (parvum, urealyticum), GBS, Gardnerella* | RT-PCR (placental tissue) | 57/57 (100%) | *U. parvum* was detected in 11 (19.3%), *G. vaginalis* in 10 (17.5%), *GBS* in 9 (15.8%), and *U. urealyticum* in 2 (3.5%) placental samples. However, *M. hominis* and *M. genitalium* were not detected. | Histological chorioamnionitis was found in 24/57 (42.1%) preterm placentas and 26/57 (45.6%) preterm placentas had at least one bacterial infection. | Only *U. parvum* was significantly associated with histological chorioamnionitis in preterm placentas (OR 5.0; 95%CI 1.2–21.5, P = 0.002). |
| | | | | *U. urealyticum* and GBS were more common in second trimester, and *G. vaginalis* was more detected in third trimester samples. | *U. urealyticum* and GBS in 2nd trimester placental sample and *G. vaginalis* in 3rd trimester were not associated with histological chorioamnionitis. | |
| Dammann, 2003 [31] | *U. urealyticum M.hominis* | Culture (placental swab) | 464/464 (100%) | *U. urealyticum* and *M. hominis* were detected in 139/464 (30%) and 27/464 (6%) of placentas, respectively. Both microorganisms were positive in 21/464 (4.5%) placental samples. | Fetal vasculitis was associated with *U. urealyticum* (53%, P ≤0.001) and *M. hominis* (14%, P ≤0.001) compared to those without fetal vasculitis. | In multivariate analysis (adjusted by gestational age, duration of membrane rupture), *U. urealyticum* culture positive placenta was associated with increased risk of fetal vasculitis (aOR 3.4; 95%CI 2.1–5.7) compared to *U. urealyticum* culture negative placenta: risk increased with a short duration of membrane rupture and with a gestational age >28 weeks. |
| Hecht, 2008 [35] | *Actinomyces spp, Prevotella spp, Corynebacterium spp, E.coli, Lactobacillus spp, staphylococcus spp, GBS, group D streptococcus, Alpha hemolytic streptococcus, anaerobic streptococcus, G. vaginalis, Mycoplasma spp, U. urealyticum* | Culture (placental tissue) | 835/835 (100%) | 41% of cultured preterm placentas were positive; *E. coli and Mycoplasma spp* were most common microorganisms. | The microorganisms associated with high grade/stage inflammation of chorionic plate and fetal vasculitis were *Actinomyces spp, Prevotella spp, Corynebacterium spp, E. coli, Lactobacillus spp, staphylococcus spp, GBS, group D streptococcus, Alpha hemolytic streptococcus, anaerobic streptococcus, G. vaginalis, Mycoplasma spp, U. urealyticum.* | In preterm birth delivered by Caesarean section, high grade/stage inflammation of the chorionic plate (OR 4.6; 95%CI 3.2–6.7), chorionic vasculitis (OR 3.9; 95%CI 2.6–5.8) and umbilical cord vasculitis (OR 3.4; 95%CI 2.2–5.4) were more associated with bacterial infection compared to placentas without inflammation. |
| Hillier, 1988 [37] | *U. urealyticum, M. hominis, G. vaginalis, Mobiluncus spp, Peptostreptococcus spp, Bateroides spp, streptococci, Lactobaccilli* | Culture (placental swab and vaginal swab) Gram stain | 38/74 (51%) | The most common microorganisms found in preterm placentas were *U. urealyticum* 18/38 (47%) and *G. vaginalis* 10/38 (26%). | Bacterial vaginosis was also associated with histological chorioamnionitis (OR 2.6, 95%CI 1.0–6.6). | After controlling for demographic and obstetric variables, preterm birth was related to bacterial infections of placentas (OR 3.8; 95% CI 1.5–9.9) and with histological chorioamnionitis (OR 5.0; 95% CI 1.6–15.3). Any microorganisms recovered from the placentas (OR 7.2; 95%CI 2.7–19.5) and *U. urealyticum* with or without other organisms (OR 9.8; 95%CI 3.5–19.5) were significantly associated with histological chorioamnionitis, regardless of gestational age. Preterm birth did not alter the result. |
| Hillier, 1991 [36] | *Genital mycoplasma (U. Urealyticum, M. hominis), Facultative bacteria (GBS, E.coli, G. vaginalis, Streptococci, Enterococcus, Lactobacillus), Anaerobic bacteria (Peptostreptococcus, Fusobacterium, Bacteroides, Actinomyces, Mobiluncus), Yeast (C. Albicans)* | Culture (placental swab) | 112/268 (42%) | Microorganisms were detected 36/112 (32%) and 29/156 (19%) placentas in cases (≤ 34 weeks' gestation) and controls (>34 weeks' gestation), respectively. Two or more bacteria were identified from 17 (15%) of 112 placentas in the case group (≤ 34 weeks' gestation) and 12 (8%) of 156 placentas in the control group (>34 weeks' gestation) (P = 0.05). | Placental histological chorioamnionitis was detected in 66/112 (59%) of cases and 35/156 (22%) of controls. | Histological chorioamnionitis and bacterial infection (*U. urealyticum* not included) were strongly associated in cases after controlling for cofounding variables e.g. mode of delivery, duration of membrane rupture and bacterial vaginosis (OR 7; 95%CI 3.0–16.4). *U. urealyticum, E. coli, Bacteriodes* were significantly associated with histological chorioamnionitis (P<0.05). *GBS* but not *Peptostreptococcus* was also associated with both preterm delivery and histological chorioamnionitis. |

(*Continued*)

**Table 2.** (Continued)

| Author, Year | Reported pathogenic microorganism | Diagnostic method | No. PTB (%) | Infection | Histopathological abnormalities | Triad of histopathology, infection and PTB |
|---|---|---|---|---|---|---|
| Honma, 2007 [38] | *U. urealyticum* | Culture (placental swab) | 105/105 (100%) | *U. urealyticum* was positive in 17 placentas (25.8%) of non-chronic lung disease (N-CLD), and 15 placentas (38.5%) of chronic lung disease group. | 39/105 (37%) preterm placentas (<32 weeks' gestation) had histological chorioamnionitis. | In multivariate analysis, histologic chorioamnionitis was associated with premature rupture of membrane (OR 10.19; 95%CI: 3.10–33.56), placental colonization of U. urealyticum (OR 6.73, 95%CI: 1.89–23.91), neonatal colonization of other microorganisms (OR 7.33, 95% CI: 1.22–44.13). |
| | | | | Other microorganisms detected in 12 placentas (18.2%) with or without *U. urealyticum* were *coagulase-negative Staphylococcus* (CoNS), *Enterococcus*, *Candida*, *α-Streptococcus*, *Enterobacter*, *Bacillus* and *Mycoplasma hominis*. | 21/39 (54%) placentas with histological chorioamnionitis had colonisation of *U. urealyticum* (OR 6.73, 95% CI:1.89–23.91). | |
| Ingrid, 2011 [39] | *Chlamydia trachomatis* | PCR (placental tissue) | 304/304 (100%) | *C. trachomatis* was harbored in 76/304 (25%) preterm placentas (≤32 weeks' gestation). | Histological evidence of placental inflammation was found in 123/304 (40%) preterm placentas: 64/123 (52%) had inflammation of maternal and fetal placental tissue, 50/123 (41%) had only maternal tissue inflammation and 4/123 (3%) had inflammation of fetal placental tissue. Other abnormal placental findings such as peripheral funisitis, acute villitis, acute intervillositis with intervillous abscesses were detected in 5/123 (4%) placentas. | *C. trachomatis* infection was more associated with placental histological inflammation: 41/76 (54%) placentas with *C. trachomatis* versus 82/228 (36%) preterm placentas without *C. trachomatis* infection (OR 2.1; 95%CI 1.2–3.5). |
| Kwak, 2014 [41] | *U. urealyticum, M. hominis* | Culture (vaginal swab) | 179/179 (100%) | 112/179 (62%) of vaginal fluid cultures were positive for genital mycoplasma (99 cases were only positive for U. urealyticum and 13 were positive for both *U. urealyticum* and *M. hominis*). But no samples were positive for *M. hominis*. | 50/179 (28%) of preterm placentas had histological chorioamnionitis. However, 36/112 (32%) of *U. urealyticum* positive cases had histological chorioamnionitis, which was statistically significant (P<0.05). | Patients with culture positive for both *U. urealyticum* and *M. hominis* 13/112 (12%) had a significantly higher proportion of preterm birth (100% (13/13) vs 73% (73/99), P<0.001) and histological chorioamnionitis (62% (8/13) vs 29% (29/99), P = 0.019) than patients only positive for *U. urealyticum*. |
| Namba, 2010 [44] | *Ureaplasma spp* | Culture (placental swab) PCR (placental tissue) | 151/151 (100%) | 63/151 (42%) of preterm placentas (<32 weeks' gestation) harboured *Ureaplasma spp* compared to 10/41 (24%) term placentas (P<0.05). | Histological chorioamnionitis was more common in preterm placentas of *Ureaplasma spp* positive (52/63, 83%) compared to placentas without infection (29/63, 46%), P<0.001. Histological chorioamnionitis with funisitis was also significantly associated with *Ureaplasma* positive preterm placentas than negative group (46% vs 7%, P<0.001). | In multivariate analysis, the association between *Ureaplasma spp* positivity in preterm placentas and histological chorioamnionitis was significant (OR 8.6; 95%CI 3.72–19.89). |
| Patel, 2018 [46] | *GBS* | Culture (vaginal and rectal swab) | 181/181 (100%) | *GBS* was positive in 55/181 (29.4%) of preterm cases. | The rate of histological chorioamnionitis was not significantly different between *GBS* negative and positive groups (81/126 (64.2%) vs 38/55 (69%), P = 0.62). Histologic chorioamnionitis was associated with earlier gestational age (29.2 versus 31.3 weeks, P < .0001) at birth. Earlier gestational age (<30 weeks) was associated with histological chorioamnionitis (P<0.001). | This single-centered, retrospective cohort study demonstrated that genital GBS colonization does not appear to be associated with an increased rate of Histologic chorioamnionitis in patients with PPROM <34 weeks of gestation. |

(*Continued*)

**Table 2.** (Continued)

| Author, Year | Reported pathogenic microorganism | Diagnostic method | No. PTB (%) | Infection | Histopathological abnormalities | Triad of histopathology, infection and PTB |
|---|---|---|---|---|---|---|
| Pettker, 2007 [47] | *U. urealyticum, Bacteriodes, Gram (+) anaerobes, E.coli, Group B Streptococcus, Peptostreptococcus spp, Streptococcus Pneumoniae* | Culture (amniocentesis microbial culture, placental tissue biopsy, placental swab) | 183/183 (100%) | Of 29/56 amniotic culture positive preterm samples, the most common micro-organism was *U. urealyticum* (12/29,31.5%). | Amniotic fluid culture negative with preterm birth was more associated with histological chorioamnionitis and funisitis than term pregnancy control group (P<0.001). | "Positive amniotic fluid cultures with preterm birth had higher severities (median grades) of funisitis and acute histologic chorioamnionitis compared with those with negative amniotic cultures and term pregnancy controls group (P<0.05)". |
| Queiros da Mota, 2013 [48] | *Bacteria species including gram positive and negative* | Culture (placental swab) | 202/376 (53.7%) | 73/376 (19.4%) placentas were bacterial culture positive; 38/73 (52%) were preterm placentas. 193 microorganisms were detected in 152 positive cultures and *gram positive cocci* was the most common. | 101/376 (26.9%) placentas had histological chorioamnionitis; 43/101 (42%) of term and 53/101 (52%) of preterm placentas. The rate of positive cultures was higher in placentas with histological chorioamnionitis compared to those without (27% vs 16%, P = 0.01). | Preterm deliveries (extremely, plus very, plus moderate, plus late preterm) in which the proportion of histologic chorioamnionitis with positive and negative cultures was 25% (14/56) and 75% (42/56) respectively. |
| Sweeney 2016 [49] | *U. parvum, U. urealyticum, GBS, Bacteroides, E. coli, G. vaginalis, Bifidobacterium spp, Propionibacterium spp* | Culture (placental swab) 16s rRNA PCR (placental tissue) | 535/535 (100%) | Microorganisms were identified in 57/535 (10.6%) placentas. A total of 61 microorganisms were isolated from these placentas and *U. parvum* (36/61, 59%) was the most prevalent. | Placentas infected by microorganisms were more likely to have histological chorioamnionitis than noninfected placentas (31/47, 54.4% vs 90/427, 18.8%; P <0.001), irrespective of gestational age and ethnicity (P = 0.528). | Histological chorioamnionitis was significantly associated with moderate/late preterm birth in the presence of *Ureaplasma spp* (P<0.001). |

Abbreviations: AC, amniocentesis; *C. albican*, candida albican; *C. trachomatis*, Chlamydia trachomatis; *E.coli*, Escherichia coli; *GBS*, group B streptococcus; *G. vaginalis*, Gardnerella vaginalis; *M. hominis*, Mycoplasma hominis; *M. genitalium*, Mycoplasma genitalium; *spp*, species; *U. urealyticum*, Ureaplasma urealyticum; *U. parvum*, Ureaplasma parvum.

was significantly increased in preterm placentas from HIV seropositive pregnant women compared to HIV negative women, and concluded that HIV infection was a risk factor for histological chorioamnionitis in preterm deliveries [33]. Feist *et al.* did not find an association between HPV and enteroviruses in the development of placental histological abnormalities (i.e., villitis of unknown origin and chronic deciduitis) in both preterm and term pregnancies [32]. Tsekoura *et al.* reported a significant association between histological signs of chorioamnionitis and an adenovirus infection in preterm placentas compared to adenovirus negative samples [50]. Other viruses commonly causing placental lesions and adverse birth outcomes include cytomegalovirus, herpes simplex virus, and parvovirus B19. However, no study investigating these viruses met the inclusion criteria of this review. More information on placental histological findings and viral infection can be seen in Table 3.

## Placental histopathology in studies of preterm birth with parasitic infection

The placental histopathological abnormality with *P. falciparum* malaria infection in pregnancy was the presence of malarial parasites and/or pigment, while no histological correlate of chorioamnionitis was described in three malaria studies [28, 40, 43]. Ategeka *et al.* [28] concluded that the risk of PTB was not increased with histological evidence of placental malaria pigment in malaria infected pregnant women from Uganda compared to those not-infected. However, placental malaria pigment is significantly associated with small for gestational age and low birth weight, and it was recommended to measure adverse birth outcomes in malaria intervention studies, especially in malaria high-transmission areas [28]. Kapisi *et al.* mentioned that

**Table 3. Identified viral infection and histopathological changes in preterm birth.**

| Author, Year | Reported pathogenic microorganism | Diagnostic method | No. PTB (%) | Infection | Histopathological abnormalities | Triad of histopathology, infection and PTB |
|---|---|---|---|---|---|---|
| Ategeka, 2019 [29] | *HIV* | Not mentioned | 18/193 (9.4%) | All women had *HIV* and treated with combination of ART for >90 days (67.4%), and were WHO *HIV* disease stage 1 (asymptomatic) (94.8%). | 102/193 (52.8%) of placentas had either maternal and/or fetal histological chorioamnionitis. 44.5% (22.5% mild, 11.0% moderate, 11.0% severe) and 28.0% (17.6% mild, 9.3% moderate, 1.0% severe) of placentas had maternal and fetal histological chorioamnionitis, respectively. | Risk of preterm birth was significantly associated with severe maternal histological chorioamnionitis compared to none-mild in HIV infected women (28.6% vs 6.0%; aOR 6.04; 95%CI 1.87–19.5, P = 0.003). Risk of preterm birth was not associated with fetal histological chorioamnionitis. |
| Feist, 2020 [32] | *HPV, enterovirus* | HPV-PCR and RT-PCR for enterovirus | 14/50 (28%) | HPV and enterovirus were not detected in any specimen with abnormal placental histopathology findings (VUE and CD). | 20 cases with VUE and 30 cases with chronic deciduitis with plasma cells were included. CD (but not VUE) was associated with PTB (4/30,15%) and PPROM (7/30, 26%). | A causal role for enterovirus and HPV in the development of VUE and CD was unlikely. |
| Gichangi, 1993 [33] | *HIV* | Enzyme immunoassay (EIA), confirmed by Western Blot | 117/467 (31.4%) | Maternal HIV-1 positive rate was 3.1% in 216 liveborn term and 8.6% in 117 preterm births. Maternal HIV-1 was independently associated with preterm birth OR 2.1 95%CI 1.1–4.0). | Preterm birth was strongly associated with histological chorioamnionitis (OR 2.3; 95% CI 1.4–3.8, P<0.001), funisitis (OR 6.7; 95%CI 3.2–14.2, P<0.001) and villitis (OR 7.8; 95%CI 12.0–35.5, P<0.001). | In HIV positive mothers, preterm placentas were associated with moderate to severe chorioamnionitis (OR 3.2; 95%CI 1.1–9.5, P<0.05) and moderate to severe funisitis (OR 6.1; 95%CI 1.2–42.7, P<0.05) compared to HIV negative preterm placentas. |
| Ladner, 1998 [42] | *HIV* | Enzyme link immunosorbent assay (ELISA) | 39/275 (14%) | 275 HIV negative and 286 HIV positive placenta were examined. The rate of STDs (24–28 weeks gestation, and all treated) was not statistically different by HIV serostatus. | Histological chorioamnionitis was not associated with serostatus: 27 (9.8%) HIV positive and 28 (9.8%) HIV negative women. No statistical association, independent of HIV serostatus, was found between histological chorioamnionitis and STDs. | In HIV positive women but not HIV negative women, the risk of preterm birth and premature rupture of membranes was higher in histological chorioamnionitis than in controls (RR 3.0; 95%CI 1.5–6.3, P = 0.003) and (RR 2.9; 95%CI 1.4–6.1, P = 0.01). |
| Ombimbo, 2019 [45] | *HIV* | Not mentioned | 81/101 (80.2%) | 38/81 (47%) preterm placentas were HIV positive cases and 43/81 (53%) were HIV negative. | Placental histopathological features including immature villi, syncytial knotting, villitis and deciduitis were not significantly different between HIV positive and negative preterm placentas. | The following placental histopathological changes between HIV positive and HIV negative preterm placentas were significantly different; fibrinoid deposition with villous degeneration (59% vs 27%, P = 0.026), syncytiotrophoplast delamination (46% vs 9%, P = 0.006), increased red cell adhesion to terminal villi (50% vs 9%, P = 0.003) and increased number of capillaries (32% vs 0%, P<0.05). |

(*Continued*)

**Table 3.** (Continued)

| Author, Year | Reported pathogenic microorganism | Diagnostic method | No. PTB (%) | Infection | Histopathological abnormalities | Triad of histopathology, infection and PTB |
|---|---|---|---|---|---|---|
| Tsekoura, 2010 [50] | *Adenovirus* | PCR (placenta tissue) | 37/58 (63.8%) | Detection of adenovirus was higher in preterm (29/71, 40.8%) than term placentas (25/122, 20.5%), (OR 2.7; 95%CI 1.4–5.1, P = 0.002). | Histological chorioamnionitis was more common in preterm than term placentas (49% vs. 19%; P = 0.025). In preterm placentas, histological chorioamnionitis was more common in adenovirus PCR-positive than adenovirus negative samples (75% vs. 36%; P = 0.026). | In adenovirus PCR-positive placentas, histological chorioamnionitis was more frequent in preterm than term placentas (75% vs. 36%; P = 0.003). However, in adenovirus PCR-negative placentas, abnormal histological findings did not differ significantly between preterm and term (36% vs. 20%; P = 0.488). |

Abbreviations: ART, anti-retroviral therapy; CD, chronic deciduitis; *HIV, human immunodeficiency virus; HPV, human papilloma virus*; OR, Odds ratio; PCR, polymerase chain reaction; VUE, villitis of unknown origin.

the risk of PTB was significantly higher if malaria parasites were detected in the placenta in comparison with non-malaria infected cases, and placental malaria pigment was not associated with preterm deliveries [40]. Lufele *et al.* also reported an association between histopathologically confirmed placental malaria infection and adverse birth outcomes. A chronic infection, defined as the presence of malaria parasites and pigment deposition in monocytes and/or fibrin, was significantly associated with increased risk of PTB, while acute placental infection, defined as the presence of malaria parasites but not pigment, increased the risk of PTB at a non-significant level [43]. Saad *et al.* investigated an association between a *Toxoplasma gondii* infection and PTB by using the mean histopathological score based on abnormalities such as fibrin deposition, intervillous haemorrhage, villous degeneration and fibrosis, haemorrhage and nearby inflammatory cellular infiltrate and infarction [34]. They reported that the highest pathological score was observed in anti-toxoplasma IgM positive cases compared to anti-toxoplasma IgM/IgG negative cases [34]. No other studies on parasites such as soil transmitted helminths, *Schistosomiasis, Trichomonas vaginalis, Giardia, Cryptosporidium* were identified for this review. The detailed information on included studies is shown in Table 4.

## Discussion

To our knowledge, this is the first systematic literature review to summarise placental histological abnormalities in PTB cases with a confirmed infection. The strength of this review is that it includes studies reporting the triad of placental histopathology, infection and PTB, allowing an inference on the causal pathways. However, the heterogeneity across studies meant it was not possible to combine the results in a meta-analysis as the included studies had different study designs, employed various diagnostic techniques to confirm causative pathogens with different test sensitivities, and used various systems for placental histopathological classification. Additionally, several PTB subcategories (spontaneous PTB with intact membranes, PPROM and indicated or induced due to maternal or fetal complications) were investigated in most of the reviewed studies, again preventing data pooling. Only 12 articles specified in the Methods section that gestational age was determined by ultrasound. Notably, only one in three studies were conducted in an LMIC, where neonatal mortality from PTB is highest [69]. Further limitations of this systematic review are that it fails to explore the association between specific causative microorganisms and specific placental histopathological changes due to study

**Table 4. Identified parasitic infection and histopathological changes in preterm birth.**

| Author, Year | Reported pathogenic microorganism | Diagnostic method | No. of PTB (%) | Infection | Histopathological abnormalities | Triad of histopathology, infection and PTB |
|---|---|---|---|---|---|---|
| Ategeka, 2020 [28] | P. falciparum | Placental blood smear, placental blood for LAMP, placental histopathology | 37/637 (5.8%) | 51.8% had microscopic parasitemia, and 82.3% had microscopic or submicroscopic parasitemia. | Women randomized to IPTp with SP had a significantly higher prevalence of malaria at delivery compared with women randomized to IPTp with DP: evidence of parasites or malaria pigment by histopathology (61.7% vs 28.2%, P< 0.001). | By binary classification, any malaria parasite or pigment detected by placental histopathology did not increase the risk of preterm birth (aRR 1.09; 95%CI 0.54–2.2, P = 0.82) but was associated with an increased risk for SGA (aRR 2.11; 95% CI 1.25–3.54, P = 0.005). By using the Bulmer classification system, acute-chronic infection (parasite and pigment detected) was not associated with PTB (aRR 1.64; 95%CI 0.47–5.76, P = 0.44), SGA (aRR 0.88; 95%CI 0.21–3.64, P = 0.86) or LBW (aRR 2.01; 95%CI 0.59–6.88, P = 0.27) compared to malaria uninfected samples. However, past-chronic infection (only malaria pigment detected) was associated with increased risk for SGA (aRR 2.14; 95%CI 1.27–3.60, P = 0.004) but not PTB (aRR 1.06; 95%CI 0.51–2.18, P = 0.88). |
| Kapisi, 2017 [40] | P. falciparum | Microscopy of placental blood smear, LAMP detection of parasite DNA in placental blood, placental histopathology | 26/282 (9.2%) | Of 282 women, 52 (18.4%) had no episodes of symptomatic malaria or asymptomatic parasitemia during the pregnancy, 157 (55.7%) had low malaria burden (0–1 episodes of symptomatic malaria and < 50% of samples LAMP+), and 73 (25.9%) had high malaria burden during pregnancy (≥ 2 episodes of symptomatic malaria or ≥ 50% of samples LAMP+). | Compared to women with no malaria exposure during pregnancy, the risk of placental malaria by histopathology was higher among low and high burden groups (aRR 3.27; 95% CI 1.32–8.12 and aRR 7.07; 95% CI 2.84–17.6). Detection of placental parasites by any method was significantly associated with PTB (aRR 5.64; 95%CI 1.46–21.8), irrespective of the level of malaria burden during pregnancy. | After adjustment (gravidity, IPTp), PTB was significantly associated with any malaria burden during pregnancy plus placental parasites detected by microscopy of placental blood smear, LAMP detection of dried placental blood spot and placental histopathology (aRR 5.88; 95%CI 1.02–34.0, P = 0.048). Risk of PTB was not significantly associated with any malaria burden during pregnancy when pigment only was detected by placental histopathology (aRR 1.74; 95%CI 0.34–8.96, P = 0.51). |

(*Continued*)

**Table 4.** (Continued)

| Author, Year | Reported pathogenic microorganism | Diagnostic method | No. of PTB (%) | Infection | Histopathological abnormalities | Triad of histopathology, infection and PTB |
|---|---|---|---|---|---|---|
| Lufele, 2017 [43] | *P. falciparum* | Peripheral blood smear (thick and thin films), placental histopathology | 81/962[a] (8.4%) | 11.9% (172/1448) experienced symptomatic malaria during their pregnancy. | Of 1451 placentas examined, 18.5% (269/1451) showed evidence of current or past PM. There were 7.5% active infections [3.7% (54/1451) acute, and 3.8% (55/1451) chronic], and 11.0% (160/1451) past infections. | After adjustment for cofounding variables, women with chronic placental malaria infection significantly had an increased risk of PTB compared to malaria uninfected women (OR 3.92; 95%CI 1.64–9.38, P = 0.002). Although acute placental malaria infection was more likely to have PTB, the finding was not significant (OR 2.33; 95%CI 0.86–6.35, P = 0.097). |
| Saad, 2017 [34] | *Toxoplasma Gondii* | Anti-*toxoplasma gondii* IgM and IgG by Indirect haemagglutination assay (sensitivity threshold 8 IU/ml) | 37/240 (15.4%) | 60 in each group defined as Group A: anti-*Toxoplasma* IgM and IgG negative; Group B: positive anti-*Toxoplasma* IgM or IgM seroconversion in mother and neonate; Group C: rising anti-*Toxoplasma* IgG during pregnancy; Group D: fixed low (non-rising) anti-*Toxoplasma* IgG during pregnancy. | The common placental histopathological findings in the IgM-positive group (group B) were fibrin deposition, intervillous haemorrhage, villous degeneration and fibrosis, haemorrhage and nearby inflammatory cellular infiltrate and infarction. Group C exhibited higher percentages of inflammatory changes then group D. | The detected abnormal placental findings were calculated and reported as the pathological mean score. This score was significantly higher (P < .001) in IgM-positive cases who ended with miscarriage or PTB (9.28 ± 0.83 and 6.80 ± 0.68, respectively) compared with IgG rising cases with the same pregnancy outcomes (4.88 ± 0.35 and 4.05 ± 0.65, respectively). |

Abbreviations: ACA, (acute) chorioamnionitis; aRR, adjusted risk ratio; BMT, basal membrane thickening; *CMV, cytomegalovirus*; *GBS, group B streptococcus*; *HPV, human papilloma virus*, IAI, intra-amniotic inflammation; LAMP, Loop-mediated isothermal amplification; MIAC, microbial invasion of the amniotic cavity; OR, odds ratio; PLMN, polymorphonuclear leuocyte; PPROM, preterm premature rupture of membranes; PRBC, parasitised red blood cell; PTB, preterm birth; SGA, small for gestational age.

[a] Total sample size was 1451 participants but only 962 patients had ultrasound scans to determine preterm birth.

diversity; and that title and abstract screening of more than 1500 manuscripts has a possibility that some studies were inadvertently overlooked.

Histological examination of the placenta is a very useful tool for assessing the multifactorial aetiology of PTB [70] as well as other adverse pregnancy outcomes (e.g., stillbirth and SGA) [71]. While placental tissue sample collection and the classification system of placental histological abnormalities varied, histological chorioamnionitis was common and strongly associated with PTB in the presence of both bacterial and viral infection in this systematic review. In malaria, histopathology can confirm infection, although it is not usually described as chorioamnionitis. The placental malaria confirmed by histopathological examination (presence of malaria infected red blood cells or malaria pigment in the intervillous space) is a better diagnostic tool for malaria infection rather than peripheral blood film due to subpatent infections and it is frequently used to assess the adverse pregnancy outcomes in malaria studies [21–23, 61, 63]. Lufele *et al.* found that chronic placental malaria infection increased the risk of PTB but the risk of PTB was not increased in patients with acute placental malaria infection [43]. Similarly, Kapisi *et al.* found that the risk of PTB was significantly higher if malaria parasites, but not pigment, were detected in the placenta. Although the association between PTB and

placental malaria remains inconclusive, it was significantly associated with other adverse birth outcomes such as SGA and miscarriage [28, 40, 43]. The only other study in the parasitic subgroup was of toxoplasmosis in which placental histopathological findings were more similar to acute and chronic inflammatory changes reported for bacteria, although the authors created their own detailed scoring system [34].

A methodological limitation in the process of diagnosing histological chorioamnionitis is that selection of a piece of placental tissue may lead to sampling of a segment with missing focal lesions in the placenta. Additionally, there is sparse evidence in humans on the time it takes for ascending intrauterine infection to result in detectable histologic chorioamnionitis which could have resulted in negative histopathology in the presence of confirmed infection [72]. It should be considered in future studies [48]. Pettker *et al.* concluded that histopathological examination of the placenta and microbiological tests fail to diagnose intra-amniotic inflammation and that caution is required in vaginal deliveries due to bacterial contamination or colonisation [47]. For an accurate interpretation of histopathological findings, clinical and procedural details should be reported. This includes exposure to antibiotics or antivirals, length of membrane rupture, number of vaginal examinations, and ascending bacterial infection. These important clinical factors were not adequately described in most of the included studies (S2 Table) and remain an area for improvement in future PTB studies.

Causality between histopathological changes in the placental tissue, PTB and infection can only be established if the causative pathogen is identified. The development of more sensitive, non-selective diagnostic tools over the last few years, provides researchers with additional options to unravel the link between the triad of PTB, infection and placental histopathological abnormalities [73]. Diagnostic tools reported in the reviewed studies included bacteriological culture, nucleic amplification techniques, immunohistochemical staining of placental tissue, with only one study using 16S rRNA sequencing [49]. Improved methods will increase the range of detected pathogens and lead to an improved understanding of gestational tissue infection. Contamination of the placenta and membranes can occur due to PPROM or during passage through the vagina, or skin during Caesarean section, and sequencing methods do not distinguish between viable and non-viable bacteria [74, 75]. Some authors in this review indicated that the role of infection in PTB may be underestimated due to limitations in the diagnostic methods [31, 47, 49], the difficulty of distinguishing between colonisation and contamination [35], and decreased bacterial abundance due to antibiotic treatment prior to tissue sampling [32, 34, 35]. No manuscript in this review reached the ideal "gold standard" to link the pathogen and placental histopathology of "amniotic fluid culture/polymerase chain reaction (PCR) and placental pathology" [70]. Only one study included amniotic fluid culture [47] and only four used a PCR test to establish the presence of microorganisms [30, 39, 44, 50].

Current consensus is that PTB is a syndrome, not a single clinical entity, that involves inherited predisposition, and maternal, fetal factors and placental factors, as well as signs of initiation of parturition and mode of delivery [76, 77]. Suggestions to improve future research on this subject include: i) adherence to internationally recognised classification of PTB subgroups; ii) adherence to standard methods of placental sampling and histopathology classification (including severity); iii) a careful approach to study design as well as sensitive and specific infection detection methods, and iv) an effort to associate placental histopathology, PTB and infection.

## Supporting information

**S1 Table. Search terms used for systematic review.**
(DOCX)

**S2 Table. Clinical characteristics associated with placental histopathology.**
(DOCX)

**S3 Table. Risk of bias assessment using Newcastle-Ottawa Scale (NOS).**
(DOCX)

**S4 Table. Placental histopathological sample collection and classification system.**
(DOCX)

**S1 Checklist.**
(DOCX)

## Acknowledgments

Special thanks to colleagues including Aung Pyae Phyo for allowing me to use a map of the included studies (under CCBY 4.0 license) and Tobias Brummaier for his helpful suggestions on this review.

## Author Contributions

**Conceptualization:** Aung Myat Min, Julie A. Simpson, Stephen H. Kennedy, François H. Nosten, Rose McGready.

**Data curation:** Aung Myat Min, Makoto Saito.

**Formal analysis:** Aung Myat Min.

**Investigation:** Aung Myat Min.

**Methodology:** Aung Myat Min, Makoto Saito, Rose McGready.

**Project administration:** Aung Myat Min.

**Resources:** Aung Myat Min.

**Software:** Aung Myat Min.

**Supervision:** Makoto Saito, Julie A. Simpson, Stephen H. Kennedy, François H. Nosten, Rose McGready.

**Validation:** Aung Myat Min, François H. Nosten, Rose McGready.

**Visualization:** Aung Myat Min.

**Writing – original draft:** Aung Myat Min.

**Writing – review & editing:** Aung Myat Min, Makoto Saito, Julie A. Simpson, Stephen H. Kennedy, François H. Nosten, Rose McGready.

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
