## [Decision Letter · Decision Letter 0]

14 Jun 2021

PONE-D-21-16155

Placental histopathology in preterm birth with confirmed maternal infection: a systematic literature review

PLOS ONE

Dear Dr. Min,

Thank you for submitting your manuscript to PLOS ONE. After careful consideration, we feel that it has merit but does not fully meet PLOS ONE’s publication criteria as it currently stands. Therefore, we invite you to submit a revised version of the manuscript that addresses the points raised during the review process.

Your manuscript has a great potential to be published but some aspects of the manuscript can be improved prior to formal acceptance. We, therefore, ask that you revise your manuscript paying close attention to the specific points detailed by Reviewer 2.  

We look forward to receiving your revised manuscript.

Kind regards,

Claudio Romero Farias Marinho, Ph.D.

Academic Editor

PLOS ONE

Journal Requirements:

2. We note that Figure 2 in your submission contain map images which may be copyrighted. All PLOS content is published under the Creative Commons Attribution License (CC BY 4.0), which means that the manuscript, images, and Supporting Information files will be freely available online, and any third party is permitted to access, download, copy, distribute, and use these materials in any way, even commercially, with proper attribution. For these reasons, we cannot publish previously copyrighted maps or satellite images created using proprietary data, such as Google software (Google Maps, Street View, and Earth). For more information, see our copyright guidelines: http://journals.plos.org/plosone/s/licenses-and-copyright.

2.1.    You may seek permission from the original copyright holder of Figure 2 to publish the content specifically under the CC BY 4.0 license. 

2.2.    If you are unable to obtain permission from the original copyright holder to publish these figures under the CC BY 4.0 license or if the copyright holder’s requirements are incompatible with the CC BY 4.0 license, please either i) remove the figure or ii) supply a replacement figure that complies with the CC BY 4.0 license. Please check copyright information on all replacement figures and update the figure caption with source information. If applicable, please specify in the figure caption text when a figure is similar but not identical to the original image and is therefore for illustrative purposes only.

Reviewers' comments:

Reviewer's Responses to Questions

**Comments to the Author**

1. Is the manuscript technically sound, and do the data support the conclusions?

Reviewer #1: Yes

Reviewer #2: Yes

2. Has the statistical analysis been performed appropriately and rigorously? 

Reviewer #1: Yes

Reviewer #2: N/A

3. Have the authors made all data underlying the findings in their manuscript fully available?

Reviewer #1: Yes

Reviewer #2: Yes

4. Is the manuscript presented in an intelligible fashion and written in standard English?

Reviewer #1: Yes

Reviewer #2: Yes

5. Review Comments to the Author

Reviewer #1: This is a systematic and sober review of studies addressing the interaction among preterm birth, histopathology, and infection. Its strength and weaknesses are detailed at the start of the Discussion section. The very rigorous application of exclusion criteria that reduced the initial set of >1,500 studies to only 23 may be of concern (were important studies missed?). A similar concern is related to the fact that only very few of the included studies were conducted in the low and middle-income countries that endure the most of the problem triad examined – but that is in the nature of systematic reviews. I have no specific comments to this manuscript, which provides a convenient resource for readers looking for a well-written, quick and informative overview.

Reviewer #2: This systematic review examines the relationship between infections, placental pathology and preterm birth. As the authors acknowledge, the studies included are quite disparate in many ways making this more of a summary of variety of rather differently designed studies, than having an overarching “story”. Not only do they cover different types of infectious agents (bacteria viruses and parasites), but these may affect the placenta via different routes (most notably blood borne vs ascending infections) and so are manifest in different parts of placental tissue (e.g. blood spaces vs membranes). Additionally, some studies include only preterm births while in others preterm births are a fraction of a larger population.

The review does appear comprehensive and contains well-crafted summaries of key characteristics of selected studies. I have only a few comments for the authors.

Major points

1. Line 64-66, quote from Redline: One area where greater clarity could be provided is the separation between blood borne causes of placental infection (most notably malaria, where CAM appears not to be a complication) and ascending infection- where the quote is more apt. Here and elsewhere a clearer separation between bacteria and malaria would be beneficial.

2. Line 304: discusses histological examination for infection but only focuses on CAM and intraamniotic infection.

Minor comments

1. Table 1 Dammann et al: what is the gestation age used? As currently written it is not clear.

2. In this table Ombimbo and Kapisi are described as being in LM countries not LMIC. The categorisaion is not very accurate- Kenya and PNG are classified as LIC while Uganda with a lower GNP is classified as LM. I would suggest all be classed as LMIC (in opposition to the HIC)

3. Table 2 Pettker last column: Please reword text, does not make sense. In column 5 U urealyticum is described as being the majority when it is (perhaps) most common but not majority.

4. Table 3 Attega: I could not readily find definitions for “maternal” vs “fetal” chorioamnionitis. Please provide descriptions of this unusual classification.

5. Gichangi column 5 : replace CU with CI

6. Table 4: why is “placental histopathology” described as a method of malaria diagnosis in one study only, when it seems it was used in all three?

7. Line 314-5: “Not enough time has passed” – this seems unlikely, any supporting evidence?

8. References: there are glitches or lacking data in references 6, 24, 35, 51, 54, 55, 57, 58, 61, 63, 66

9. Table S4 column 3, Ombimbo, the last column text is partly scrambled.

10. PRISMA: Points 25 and 26 refer to “in another document” but I did not see this.

6. PLOS authors have the option to publish the peer review history of their article (what does this mean?). If published, this will include your full peer review and any attached files.

Reviewer #1: No

Reviewer #2: No

---

## [Author Response · Author response to Decision Letter 0]

22 Jul 2021

The manuscript has been reviewed to make sure it complies with PLOS ONE formatting style and author titles/affiliations.

The referencing style has also been changed from Vancouver superscript to ordinary Vancouver.

2. We note that Figure 2 in your submission contain map images which may be copyrighted. All PLOS content is published under the Creative Commons Attribution License (CC BY 4.0), which means that the manuscript, images, and Supporting Information files will be freely available online, and any third party is permitted to access, download, copy, distribute, and use these materials in any way, even commercially, with proper attribution. For these reasons, we cannot publish previously copyrighted maps or satellite images created using proprietary data, such as Google software (Google Maps, Street View, and Earth). For more information, see our copyright guidelines: http://journals.plos.org/plosone/s/licenses-and-copyright. We require you to either (1) present written permission from the copyright holder to publish these figures specifically under the CC BY 4.0 license, or (2) remove the figures from your submission:

2.1. You may seek permission from the original copyright holder of Figure 2 to publish the content specifically under the CC BY 4.0 license. We recommend that you contact the original copyright holder with the Content Permission Form (http://journals.plos.org/plosone/s/file?id=7c09/content-permission-form.pdf) and the following text:

Please upload the completed Content Permission Form or other proof of granted permissions as an "Other" file with your submission.In the figure caption of the copyrighted figure, please include the following text: “Reprinted from [ref] under a CC BY license, with permission from [name of publisher], original copyright [original copyright year].”

 The original map creator has been contacted and signed the CCAL form which has been attached.

5. Review Comments to the Author

Reviewer #1: This is a systematic and sober review of studies addressing the interaction among preterm birth, histopathology, and infection. Its strength and weaknesses are detailed at the start of the Discussion section. The very rigorous application of exclusion criteria that reduced the initial set of >1,500 studies to only 23 may be of concern (were important studies missed?). A similar concern is related to the fact that only very few of the included studies were conducted in the low and middle-income countries that endure the most of the problem triad examined – but that is in the nature of systematic reviews. I have no specific comments to this manuscript, which provides a convenient resource for readers looking for a well-written, quick and informative overview.

Thank you. 

It is true that most of 1529 articles were excluded based on the information in the title and abstract and there is a possibility some studies may have been missed and we have included as the statement in the limitation (line 322-323). Of the 122 articles that were eligible for full text screening only 23 were passed by two independent reviewers. 

Reviewer #2: This systematic review examines the relationship between infections, placental pathology and preterm birth. As the authors acknowledge, the studies included are quite disparate in many ways making this more of a summary of variety of rather differently designed studies, than having an overarching “story”. Not only do they cover different types of infectious agents (bacteria viruses and parasites), but these may affect the placenta via different routes (most notably blood borne vs ascending infections) and so are manifest in different parts of placental tissue (e.g. blood spaces vs membranes). Additionally, some studies include only preterm births while in others preterm births are a fraction of a larger population.

The review does appear comprehensive and contains well-crafted summaries of key characteristics of selected studies. I have only a few comments for the authors.

Thank you.

Major points

1. Line 64-66, quote from Redline: One area where greater clarity could be provided is the separation between blood borne causes of placental infection (most notably malaria, where CAM appears not to be a complication) and ascending infection- where the quote is more apt. Here and elsewhere a clearer separation between bacteria and malaria would be beneficial.

Thank you for the comments. This has been addressed in two places in the introduction (Line number: 67-70, 88-95)

2. Line 304: discusses histological examination for infection but only focuses on CAM and intraamniotic infection.

We agree and the discussion has now been extended beyond CAM and intraamniotic infection (Line number: 329-342)

Minor comments

1. Table 1 Dammann et al: what is the gestation age used? As currently written it is not clear.

The gestational age used in Dammann et al. was not clearly defined. The information is only found in Table 2, which mentioned the gestational age as <26, 26-28, and >28 week, so this has now been amended in Table 1: as “not clearly defined”.

2. In this table Ombimbo and Kapisi are described as being in LM countries not LMIC. The categorisaion is not very accurate- Kenya and PNG are classified as LIC while Uganda with a lower GNP is classified as LM. I would suggest all be classed as LMIC (in opposition to the HIC)

I agreed with your suggestion and change all LIC to LMIC.

3. Table 2 Pettker last column: Please reword text, does not make sense. In column 5 U urealyticum is described as being the majority when it is (perhaps) most common but not majority.

Thank you. Amended.

4. Table 3 Attega: I could not readily find definitions for “maternal” vs “fetal” chorioamnionitis. Please provide descriptions of this unusual classification.

Thank you. This can be found The definition for “Maternal” vs “Fetal” chorioamnionitis in supporting table 4B. Redline classification system was used to define histologic chorioamnionitis including maternal and fetal inflammatory response of the placenta in Ategeka, 2019.

5. Gichangi column 5 : replace CU with CI

Thank you. Amended (good eyes!).

6. Table 4: why is “placental histopathology” described as a method of malaria diagnosis in one study only, when it seems it was used in all three?

Amended.

7. Line 314-5: “Not enough time has passed” – this seems unlikely, any supporting evidence?

Thank you. We have amended the sentence and provided a reference (line 345-348).

8. References: there are glitches or lacking data in references 6, 24, 35, 51, 54, 55, 57, 58, 61, 63, 66

Thank you. This has been corrected. 

9. Table S4 column 3, Obimbo, the last column text is partly scrambled.

The sentence has been rewritten.

10. PRISMA: Points 25 and 26 refer to “in another document” but I did not see this.

Sorry, this was an error. The points are now addressed here: 

Point 25: Describe sources of financial or non-financial support for the review, and the role of the funders or sponsors in the review.

This research was funded in whole, or in part, by the Wellcome Trust [220211]. For the purpose of Open Access, the author has applied a CC BY public copyright license to any Author Accepted Manuscript version arising from this submission.

Point 26: Declare any competing interests of review authors.

The authors declare no competing interests.

---

## [Editor Report · Decision Letter 1]

27 Jul 2021

Placental histopathology in preterm birth with confirmed maternal infection: a systematic literature review

PONE-D-21-16155R1

Dear Dr. Aung Myat Min,

We’re pleased to inform you that your manuscript has been judged scientifically suitable for publication and will be formally accepted for publication once it meets all outstanding technical requirements.

Kind regards,

Claudio Romero Farias Marinho, Ph.D.

Academic Editor

PLOS ONE
---

## [Editor Report · Acceptance letter]

5 Aug 2021

PONE-D-21-16155R1 

Placental histopathology in preterm birth with confirmed maternal infection: a systematic literature review 

Dear Dr. Min:

I'm pleased to inform you that your manuscript has been deemed suitable for publication in PLOS ONE. Congratulations! Your manuscript is now with our production department. 

Kind regards, 

on behalf of

Dr. Claudio Romero Farias Marinho 

Academic Editor

PLOS ONE